# Structural Analysis of *Plasmodium falciparum* Hexokinase Provides Novel Information about Catalysis Due to a *Plasmodium*-Specific Insertion

**DOI:** 10.3390/ijms241612739

**Published:** 2023-08-13

**Authors:** Melissa Dillenberger, Anke-Dorothee Werner, Ann-Sophie Velten, Stefan Rahlfs, Katja Becker, Karin Fritz-Wolf

**Affiliations:** 1Biochemistry and Molecular Biology, Interdisciplinary Research Center, Justus Liebig University, D-35392 Giessen, Germany; melissa.dillenberger@uni-marburg.de (M.D.);; 2Institute of Virology, University of Marburg, Hans-Meerwein-Str. 2, D-35043 Marburg, Germany; 3Max-Planck Institute for Medical Research, Jahnstr. 29, D-69120 Heidelberg, Germany

**Keywords:** crystal structure, glycolysis, hexokinase, malaria, posttranslational modification, redox regulation

## Abstract

The protozoan parasite *Plasmodium falciparum* is the causative pathogen of the most severe form of malaria, for which novel strategies for treatment are urgently required. The primary energy supply for intraerythrocytic stages of *Plasmodium* is the production of ATP via glycolysis. Due to the parasite’s strong dependence on this pathway and the significant structural differences of its glycolytic enzymes compared to its human counterpart, glycolysis is considered a potential drug target. In this study, we provide the first three-dimensional protein structure of *P. falciparum* hexokinase (*Pf*HK) containing novel information about the mechanisms of *Pf*HK. We identified for the first time a *Plasmodium*-specific insertion that lines the active site. Moreover, we propose that this insertion plays a role in ATP binding. Residues of the insertion further seem to affect the tetrameric interface and therefore suggest a special way of communication among the different monomers. In addition, we confirmed that *Pf*HK is targeted and affected by oxidative posttranslational modifications (oxPTMs). Both S-glutathionylation and S-nitrosation revealed an inhibitory effect on the enzymatic activity of *Pf*HK.

## 1. Introduction

Malaria is an infectious disease that threatens global health and life. In 2021, the World Health Organization reported 247 million cases of malaria infection worldwide, leading to 619,000 deaths [1]. These numbers and the possible imminent resistance of the disease against currently used drugs underscore the urgency to develop new strategies for antimalarial treatment [2,3]. Malaria is caused by the protozoan parasite *Plasmodium*. Among the five human pathogen species, infections with *P. falciparum* are responsible for the most fatal cases and the highest mortality [1]. Symptoms occur during the infection of erythrocytes and are mainly caused by cytoadherence. The parasite’s replication within erythrocytes requires high levels of glucose as an energy supply to produce ATP via glycolysis [4,5,6]. Therefore, glycolysis is not only enhanced but indispensable for the survival of intraerythrocytic stages of *Plasmodium* and has attracted attention in the context of potential drug development [7,8].

*Pf*HK is the first and simultaneously rate-limiting enzyme of glycolysis, catalyzing the phosphorylation of glucose to glucose-6-phosphate (G6P). ATP provides the phosphoryl group and the reaction requires divalent cations, usually Mg^2+^ [9,10,11]. The production of G6P is not only important for glycolytic flux but is also crucial and rate-limiting for the pentose phosphate pathway and the supply of NADPH as a reducing agent, which G6P dehydrogenase provides [12,13]. *Pf*HK exists as a single-copy gene on chromosome 6 without any isoenzymes [14]. Hexokinases are well conserved among *Plasmodium*, but the sequence identity to their human counterpart is lower than 32% [7,15]. Compared to other proteins within similar species, *Pf*HK contains 15 cysteines (~3%) and is therefore considered to be exceptionally cysteine-rich [16]. Cysteines provide the basis for redox regulation via oxidative posttranslational modifications (oxPTM), which has already been reported to occur on various glycolytic enzymes. Previous studies suggested that S-glutathionylation and S-nitrosation target *Pf*HK [17,18].

Within this study, we provide the first three-dimensional structure of *Pf*HK with a 2.8 Å resolution solved via X-ray crystallography. Both glucose and citrate were identified as ligands bound to conserved residues of the active site. The structure provides novel information about the closure of the active site and the interaction of the single monomers within the tetrameric assembly of *Pf*HK and thus allows for a better understanding of its oligomerization behavior, which is quite distinct from hexokinases of other organisms. Moreover, we confirmed that S-glutathionylation and S-nitrosation both target and inhibit the enzyme.

## 2. Results

### 2.1. Overall Structure of P. falciparum Hexokinase

We obtained trigonal crystals of *Pf*HK with a resolution of 2.8 Å. All structures were solved via molecular replacement. Data collection and refinement statistics are summarized in Table 1. The crystals belonged to space group P3_1_21 and contained two monomers within the asymmetric unit (Figure 1A). The overall fold of each monomer is similar to that of other hexokinases and consists of two domains: a small domain comprising residues E94-L239 and a large domain comprising residues M1-Q93 and N240-P490, with the active site located between both domains. The binding sites for the substrates’ glucose and ATP are highly conserved among different species. The active site of both monomers contained a glucose molecule and, in one monomer, a citrate molecule was additionally bound. The two monomers are essentially similar (the RMSD is 0.7 Å with 463 residues). However, the superimposition of both monomers revealed the movement of two loops. Those loops, G104-F108 (catalytic loop; C-loop) and L130-G141 (*Plasmodium*-specific insertion; P-insert) are located in the small domain within the cleft of the substrate-binding site and will be described in more detail in the following sections (Figure 1B). The average B-factor of the structure was relatively high at 90 Å^2^, but this was to be expected since the Wilson B-factor of the data was also high (96 Å^2^). In monomer A, the average B-factor (103 Å^2^) for the P-insert region was in the same range as the average B-factor of the whole structure but was moderately increased in subunit B (124 Å^2^). These findings are in line with the electron density and indicate some flexibility.

### 2.2. Glucose and Citrate Were Bound to the Active Site

Hexokinases catalyze the phosphorylation of glucose to G6P. For this purpose, ATP serves as a phosphoryl donor and, after catalysis, both products (G6P and ADP) are released. Previous X-ray crystallographic studies have demonstrated that the enzyme exhibits two conformational states: the open state, which occurs prior to glucose binding, and the closed state, where the two lobes surrounding the glucose substrate are closed upon glucose binding [19,20]. In both subunits of the present structure, glucose is coordinated by strictly conserved residues from both domains: S183, P185, T200, K201, N240, D241, I265, G269, E292, Q319, and E322 (Figure 2A). Accordingly, a comparison with the structures of the open and closed form of *P. vivax* hexokinase (*Pv*HK, PDB ID: **6VYF, 6VYG**) showed that this *Pf*HK structure corresponds to the closed form of *Pv*HK with an RMSD of 1.1 Å for 413 residues. In addition to glucose, a citrate molecule was bound in one subunit (A) of *Pf*HK. The citrate interacts with conserved residues from both domains: G104, N107, R109, T268, and S436, which are typically involved in ADP/ATP binding (Figure 2B).

Human hexokinase-I (PDB ID: **1DGK**) is a pseudo dimer whose subdomains are connected by a helix (residues 448–474). Only in the active site of the second subdomain (475–913), glucose and an ADP molecule were bound, which is why we used this part for the comparison. The superposition revealed an RMSD of 1.6 Å (405 residues, 33.4% sequence identity). As described for the glucose binding site, the residues responsible for ADP/ATP binding are also highly conserved among all organisms. In *Pf*HK, the putative ADP/ATP binding site consists of loop residues G104-N107 and β-turn residues G267-T268, both interacting with the β- and γ-phosphates of ADP/ATP. The adenine ring would be sandwiched between residues N357-S362 and L437-W443 (Figure 2C).

Superimposition with human hexokinase structures complexed with the products—G6P (**1HKB**) or ADP (**1DGK**)—revealed that the phosphates of ADP or G6P occupy the same binding site as the citrate molecule in *Pf*HK (Figure 2C). Therefore, the citrate is located at the site where phosphate is normally transferred from ATP to glucose. Moreover, it interacts with residues used in homologous hexokinase structures to bind ATP/ADP. These findings could be further confirmed with other hexokinase structures with bound G6P or ADP (PDB IDs: **1BG3**, **6JJ9**).

### 2.3. A Plasmodium-Specific Insertion (P-Insert) Lined the Active Site

A structural comparison of the two *Pf*HK subunits and human hexokinase (PDB ID: **1DGK**) revealed that the connection between a conserved β-strand and a conserved α-helix is different. In the human crystal structure, this connection consists of residues P115-S122, whereas in *Plasmodium*, it comprises residues K131-T148 (Figure 3A). However, the most significant differences are observed in the region of the mobile loop L130-G141 (P-insert) in *Pf*HK, which, in humans, involves only three residues (P115-N117). *Pf*HK shares 89.6% sequence identity with *Pv*HK and sequence alignment has shown that this insertion is also present in *Plasmodium vivax*, suggesting a *Plasmodium*-specific insertion [15]. In both the open and closed forms of the *Pv*HK structures (PDB IDs: **6VYF**, **6VYG**), coordinates were not deposited for regions K131-K145, F167-N177, and V227-V235, nor for any ligands. This indicates that the P-insert loop was disordered in the *Pv*HK structures.

While the P-insert is most probably a *Plasmodium*-specific insertion, the hexokinase from *Eimeria tenella* (*Et*HK, PDB IDs: **6KSR**, **6KSJ**) possesses a similar insertion, though this insertion is three amino acids shorter (Figure 3B). *E. tenella* is another apicomplexan parasite with a sequence similarity of 43% compared to *Pf*HK. Both *Et*HK structures are in the open state, although one (PDB ID: **6KSR**) has a galactose molecule bound in the active site. There is no indication whether bound glucose instead of galactose would result in a close conformation. However, since this *Pf*HK structure is in the closed conformation, superposition with the open conformation of *Et*HK (PDB ID: **6KSR**, RMSD of 2.2 Å at 396 residues) showed a large shift of the small domain compared to *Pf*HK. Consequently, the different structural arrangement of the “P-insert” was not comparable.

### 2.4. Does Catalysis Affect the Conformation of the C-Loop and the P-Insert?

As mentioned above, the superimposition of both *Pf*HK monomers revealed different conformations for the C-loop (G104-F108) and the P-insert (L130-G141, Figure 4). Except for four residues in subunit B (G134-S137), the electron densities of both the P-insert and the C-loop are well-defined in both monomers. Since the chain trace of regions L130-T133 and H138-G141 is very different in the two subunits, the conformation of the linking residues must also be different, thus extending over the entire P-insert.

In monomer B, P-insert residues L130-G141 adopt a completely different conformation compared to monomer A, pointing into the active site channel and resulting in a large shift of up to 20 Å for residues G134-E139 (Appendix A). In this conformation, S135 forms a hydrogen bond to loop residue D354; furthermore, S135 is now in van der Waals distance to Y329. P-insert residues T133-Y136 interact with Y312 and S353-N357 from the large domain (Figure 4). Interestingly, some of these residues are part of the interface between the two subunits, which is described in more detail in Section 2.6 (Oligomerization of *Pf*HK).

In subunit A, all these contacts are lost, and access to the active site is broadened by the rearrangement of the conserved C-loop and P-insert. Compared to subunit B, C-loop residue T106 is shifted by 6 Å. To be sure that these observed conformational changes are induced by citrate and not by the crystal packing, we analyzed the crystal contacts. There are no contacts between the C-loop and symmetry-equivalent monomers in either subunit (Figure 1). In subunit A, P-insert residues G134-Y136 interacted via a few hydrophobic interactions with a neighboring tetramer, more specifically, with residues N87-P90, Y275, K287, and C260-Y261 of subunit B (Appendix A). In subunit B, there are no interactions of the P-insert loop with neighboring molecules, and there is enough space for the P-insert to adopt the same conformation as in subunit A. Thus, the P-insert can move freely, making the active site accessible to the solvent and, in principle, allowing a citrate molecule to bind.

### 2.5. Structural Comparisons Suggest a Plasmodium-Specific Mechanism

The superimposition of the two subunits (A and B) revealed that the position of the citrate molecule from subunit A would collide with the C-loop in subunit B, where the C-loop and P-insert are facing the inside of the active site (Figure 1, Figure 3 and Figure 4). Residue S436—which was shown to form a hydrogen bond with the citrate molecule—and residue S362 are conserved residues of the ATP/ADP binding pocket (Figure 2C). The binding of citrate requires the movement of C-loop residues (G105-N107) and, subsequently, a rearrangement of the P-insert. This movement of the C-loop can also be observed in the structures of human glucokinase. Superimposition with human glucokinase (PDB ID: **3VEY** [21]) in complex with glucose and ATPγS, which mimics a quasi-transition state, showed that the binding of ATP causes the C-loop to adopt the same conformation as in the *Pf*HK monomer A with bound citrate (Figure 4B). In the related structure (PDB ID: **3VEV**) with solely bound glucose, the C-loop adopts the same conformation as in subunit B (no citrate) from *Pf*HK (Figure 4C). Since only glucose is bound, the C-loop adopts a pre-catalytic conformation. However, the same conformation can also be observed in structures containing the products ADP (PDB ID: **1DGK**, Figure 2C) or G6P (PDB ID: **1HKB**). Thus, the conformation of the C-loop adopts the same conformation before (pre-catalytic, PDB ID: **3VEY**) and after (post-catalytic, PDB IDs: **1DGK, 1HKB**) catalysis.

As mentioned before, the citrate molecule was bound to the phosphate binding and transfer site of subunit A. Thus, it forced a rearrangement of the C-loop in this subunit, followed by a rearrangement of the P-insert, resulting in a catalytically active conformation of both loops. In our structure, the bound citrate could mimic the mechanism upon ATP binding, as observed in the quasi-transition state structure of human glucokinase (PDB ID: **3VEY**) in complex with glucose and ATPγS. In subunit B, the P-insert occupied the ATP-binding site via interactions with the large domain, thus inhibiting ATP binding and causing the C-loop and P-insert to adopt a pre- or post-catalytic conformation. The conformation of the P-insert in subunit B would hinder product release, but the rearrangement of the P-insert in subunit A would facilitate the release of the products (Figure 5). This could be a *Plasmodium*-specific hexokinase mechanism that potentially affects the enzyme’s activity.

### 2.6. Oligomerization of P. falciparum Hexokinase

The asymmetric unit of the crystal contains a dimer (AB), forming a tetramer together with the symmetry-equivalent dimer (A′B′). The two dimers are connected by a 2-fold rotation axis (Figure 6A). In accordance with that, size-exclusion chromatography showed that *Pf*HK assembles as a tetramer (Figure 6B,C). All enzymatic activities were measured with tetrameric *Pf*HK without any reducing agent, unless described otherwise.

Due to the crystallographic symmetry relationship between the dimers, two interfaces are identical in each case so that there are only four different interfaces AB (≙A′B′), AB′ (≙BA′), AA′, and BB′, which are shown in Figure 7A. Analysis with the PISA server (https://www.ebi.ac.uk/pdbe/pisa/, accessed on 11 April 2022) revealed that the subunits are tightly connected via hydrogen bonds and hydrophobic interactions.

The interface AB involves mainly residues from the large domains of subunits A and B. They are strongly connected via seven different hydrogen bonds and one additional salt bridge between E28^A^ and K348^B^. Moreover, interactions among the residues T302-Y312, R336-K348, and W351 from both monomers could be observed. In monomer B, residues K35-Q39 participate, whereas, in monomer A, residues R32-Q39 form the interface region (Figure 7B).

Residues from the small domains mainly form the AA′ or BB′ interface (Figure 7C). Subunits A and B are quite similar (RMSD 0.7 Å at 463 residues), and the differences between the subunits occur mainly in the P-insert and C-loop regions, neither of which is directly involved in the AA′ and BB′ interfaces. Thus, we suppose that the interfaces AA′ and BB′ are quite similar. Indeed, the PISA analysis revealed that there are only minor differences between the interfaces. There are six hydrogen bonds in both interfaces, mainly between N70 and the region around I197, in AA′ to L196 and I197, and in BB′ to I197, D198, and E214; in AA, there is an additional hydrogen bond between R207 (O) and K283 (NZ). The interface residues are K67-E76, K80, T187-D198, R207-K220, K283, and R316-R318.

In contrast to the multiple hydrogen bonds of interfaces AB and AA′, subunits A and B′ are mainly connected with hydrophobic interactions between a long helix (E49-R66) from both subunits. Further connection is mediated via a hydrogen bond between N53^A^ and Y56^B′^, interactions between residues K281-Y282, D299, I423 from monomer A, and residues S45-K67 and K298-D299 from monomer B′. Interestingly, H75 from all four monomers assemble within the center of the tetramer and are part of the tetrameric interfaces (Figure 7D). Distances between the H75 sidechains (atom NE) are ~3.6 Å between subunit B and B′ or A and A′ and 5.9 Å between subunits A (CE1), B (NE), or subunits A′ and B′.

As already discussed by Srivastava and colleagues, the tetrameric assembly of *Plasmodium* hexokinases is unique since most hexokinases occur in a biologically relevant manner as monomers or (pseudo-)dimers. *Pv*HK shares 89.6% sequence identity with the present *Pf*HK [15]. The structures (PDB IDs: **6VYG** and **6VYF**) of *Pv*HK were solved via cryo-electron microscopy at a resolution of 3.3 Å and 3.5 Å, respectively, and the authors supposed a bound glucose or other similar sugar molecule in its active site, but the deposited PDB file does not contain any ligands. Superimposition of the dimers resulted in 1.8 Å with 824 residues. However, the *Pv*HK structure regions K131–K145, F167–N177, and V227–V235 are not defined. A comparison of both *Plasmodium* tetramers showed differences in the interfaces of AA′ and AB that involve the small domain (2.2 Å with 1633 residues). As described above, the P-insert is rearranged in our structure upon citrate binding and probably also upon ATP binding. This movement is responsible for the loss of multiple contacts between the small and large domains in monomer A. Of particular importance appears to be the interaction between the P-insert residues L130-S135 (small domain) and region P304-Y312 (large domain), which, in turn, is part of the AB interface (Figure 4 and Figure 7B). The loss of this contact alters the AB interface and affects the neighboring molecule in the tetramer. This could be an allosteric effect, possibly explaining the binding of citrate (or physiologically ATP) only to one subunit.

### 2.7. Oxidative Posttranslational Modifications Inhibit the Enzymatic Activity of PfHK

Redox regulation via oxPTMs occurs under both physiological and pathophysiological conditions at the protein’s cysteine residues and can have a major influence on protein conformation, stability, function, and interactions with other proteins [22,23,24]. OxPTMs target hexokinases from a variety of organisms. Previous studies gained important insights into the patterns of S-glutathionylation, S-nitrosation, and S-sulfenylation within *P. falciparum* trophozoite stages and indicated that *Pf*HK was a target of S-glutathionylation and S-nitrosation under both basal conditions and conditions with increased oxidative stress [17,18,25]. A prerequisite for those modifications is the presence of reactive thiols. Compared to other proteins within similar species, *Pf*HK contains 15 cysteines (3%) and is therefore considered to be exceptionally cysteine-rich [16]. S-glutathionylation of purified, recombinant *Pf*HK was shown via α-glutathione Western blot analysis, whereas S-nitrosation was confirmed with a biotin-switch assay followed by α-biotin Western blot analysis, described in more detail in *Methods* (Figure 8A,B).

To investigate the possible effects of those modifications on the enzymatic activity of purified recombinant *Pf*HK, we performed kinetic analyses of modified enzymes. To ensure adequate recombinant production of *Pf*HK prior to our analyses, we measured K*_M_* values for comparison with former studies. The K*_M_* value for ATP was 0.83 ± 0.08 mM and it was 0.13 ± 0.01 mM for glucose. Interestingly, the K*_M_* for glucose was almost 5-fold lower when comparing it to previously measured recombinant *Pf*HK Michaelis-Menten constants [7].

S-Glutathionylation was induced by incubating *Pf*HK with different concentrations of oxidized glutathione (GSSG, 0–10 mM). The enzymatic activity decreased in a dose-dependent manner, and we observed a time-dependent inhibition of *Pf*HK activity. After 30 min of incubation, its enzymatic activity was reduced by more than 70% (Figure 8C,D). To induce S-nitrosation, *Pf*HK was incubated with varying concentrations of nitroso-cysteine (Cys-NO, 0–2 mM). Comparable to the effects observed for S-glutathionylation, S-nitrosation inhibited the enzymatic activity of *Pf*HK in a dose- and time-dependent manner (Figure 8E,F). The incubation with Cys-NO reduced its activity by ~40% after 30 min. The inhibitory effect of both modifications could be reversed by adding DTT, which was also shown to remove the signal in anti-glutathione Western blot analysis (Figure 8G).

Theoretically, *Pf*HK possesses 15 cysteines that can be oxidized. By using Ellman’s reagent, we identified 10–11 cysteines susceptible to oxidation of their sulfur group. This is in accordance with the MS analysis of modified *Pf*HK. From the 15 present cysteines, not all were detected via MS; however, 12 were found to be S-nitrosated. Six out of those modified cysteines were found to be also susceptible to S-glutathionylation, whereby a specific modification pattern could not be determined (Appendix A). The present three-dimensional structure further supports these findings: within the tetrameric assembly, putatively, 11 cysteines are accessible for oxPTMs. According to this structure, none of the cysteines are located in the active site; therefore, modified cysteines could only indirectly affect the enzymatic activity. C186 and C273 are located adjacent to the glucose binding site (7.5 and 9 Å apart, respectively), but their accessibility for oxPTMs could not be confirmed. The modification of C77 and C85 could induce conformational changes in beta sheets, potentially resulting in decreased enzymatic activity. Due to a reductant in the crystallization buffer, residues C249 and C260 are reduced in our structure, but they could easily form a disulfide (Appendix A).

## 3. Discussion

In this study, we present the first *Pf*HK structure in a closed-state conformation complexed with glucose. Our structure provides novel information about the small domain involved in the unique tetrameric assembly of *Plasmodium* hexokinases. Among the two monomers within the asymmetric unit, we observed conformational changes in the small domain of monomer A, induced by binding a citrate molecule. The residues involved are found in a *Plasmodium*-specific insertion (P-insert) and give insights into a so-far unknown mechanism: the binding of citrate seems to trigger a quasi-transition state, similar to the state observed in the structure of human glucokinase in complexes with glucose and ATPγS (PDB ID: **3VEY**), which shows that the binding of ATP causes the catalytic loop to adopt the same conformation as in monomer A with bound citrate (Figure 4B). Additionally, *Pf*HK is targeted by oxPTMs, more precisely, by S-glutathionylation and S-nitrosation, which both inhibit enzymatic activity. Based on the structural analyses, the susceptibility of individual cysteines to these modifications is discussed.

In mammalian tissues, four distinct hexokinases have been identified, referred to as type I, II, III, and IV. Mammalian hexokinase type IV is usually termed ‘glucokinase’. While types I-III have a size of ~100 kDa and are believed to have evolved from duplication of the original hexokinase gene, mammalian glucokinase (type IV) is a ~50 kDa isoenzyme [26]. For yeast, two isoenzymes were reported, both being similar to mammalian glucokinase [20]. The domain organization observed in our structures is well conserved and was previously reported for various organisms [19,27,28]. Among different *Plasmodium* species, the sequence identities to *P. falciparum* cover a range from 85.5% (*P. relictum*) to 99.8% (*P. reichenowi*) identity (analyzed with NCBI Smart BLAST). Since *Pv*HK (PDB IDs: **6VYG**, **6VYF**) and *Pf*HK share high sequence and structure identities, many of the interface residues reported within this study confirmed the data presented previously for the *Pv*HK tetramer [15]. However, some loops in *Pv*HK were not defined, including the P-insert (PDB IDs: **6VYG**, **6VYF**); thus, this region could not be used for comparison with our data.

Hexokinases consist of two lobes and, upon binding glucose, the lobes move toward each other. During this process, one lobe rotates 12 degrees relative to the other, resulting in movements of the polypeptide backbone by 8 Å [19]. Binding with glucose induces the closure of the cleft between the small and large domains, which envelopes the glucose molecule except for the hydroxyl group on the C6 atom, to which a phosphoryl group is subsequently transferred from ATP [11]. After catalysis, a movement of the small domain allows ADP to dissociate from the active site, followed by the release of G6P, which is the preferred order [27,29]. This is consistent with our finding that the P-insert can adopt different conformations, depending on the binding or absence of ligands at the phosphate binding site. *Pf*HK is known to be inhibited by its products ADP and G6P up to 94% and also by high concentrations of its substrate ATP [7]. The residues participating in glucose binding are conserved among different species, involving amino acids from both domains [20,27,28,30]. In contrast to other studies, Liu and colleagues suggested an alternation between different conformational states to ensure efficient substrate binding and product release. Furthermore, they postulated that binding glucose does not inevitably lead to a closed conformation but also induces an active open state [21].

Within this study, we could identify glucose and citrate as ligands within the active site of *Pf*HK. The citrate molecule, only bound in monomer A, interacts via hydrogen bonds with residues that are typically involved in phosphate and/or ATP binding [27,31,32]. We suppose that in this study, the presence of 300 mM citrate in the crystallization buffer prevented ATP (10 mM in a protein solution, further diluted by crystallization buffer) from binding, as well as the subsequent phosphate transfer to glucose, which could be the reason why we did not observe an electron density for ATP. Supplying the enzyme solution with both substrates upon crystallization certainly is an uncommon approach since one would expect catalysis to lead to an unstable enzyme and the failure of crystallization. We tried various combinations of ligands in the enzyme solutions, including ATP analogs, ADP, and G6P, without success in achieving crystal growth. Only the combination of glucose and ATP yielded crystals.

Not much is known about the regulatory impact of citrate on hexokinases: there are only scattered studies suggesting that citrate activates mammalian and yeast HK, and, to the best of our knowledge, citrate is reported to inhibit noncompetitively only one hexokinase, namely, from *Aspergillus niger* [33,34,35,36]. Nevertheless, the phosphate binding site is well conserved, which conflicts with common criteria of promising drug targets and excludes citrate as a candidate for inhibitor design. Moreover, the high citrate concentration in the crystallization buffer supports the hypothesis of its random binding rather than having a physiological significance. We hypothesize the citrate-induced state of monomer A to mimic the hexokinase conformation upon the binding of ATP. The new position of the catalytic loop (G104-F108; Figure 4A) in monomer A forces the P-insert (L130-G141) to rearrange. The movement of both loops seems to be ATP-specific since in structures complexed with ADP, G6P, or glucose, the corresponding catalytic loop adopts the pre-catalytic state (Figure 4A–C).

Hexokinases I-III are active as monomers, consisting of N- and C-terminal halves connected via a transition helix, forming a pseudo-dimer. [27,37]. For yeast hexokinases, both monomers and dimers have been reported and *Sulfulobus tokodaii* hexokinase also exists as a dimer, whereby the subunits are closely connected and therefore differ from mammalian HK pseudo-dimers [20,30,38,39]. In type I and III hexokinases, the C-termini are catalytically relevant, whereas the N-terminal halves fulfill a regulatory role. Both termini are known to communicate via conformational changes. Upon the ligation of G6P at the N-terminal half, for example, conformational changes lead to the inhibition of catalysis at the C-terminus via an allosteric mechanism [27,32,37,40,41]. We hypothesize that the two different conformational changes within the *Pf*HK subunits adopt a similar way of communication among the different monomers, probably mediated via displaced interface residues. This could explain how *Pf*HK is regulated upon catalysis, in addition to switching between the open and the closed conformation: The subunits putatively alternate substrate binding, which could be why citrate was bound only in one subunit of the present dimer. In subunit B, where solely glucose was bound, residues L130-S135 (small domain), which are part of the P-insert, are in contact with regions P304-Y312 (large domain), which, in turn, are part of the interface AB (Figure 4 and Figure 7B). In monomer A with bound citrate, this contact is lost, resulting in an altered AB interface and thus also affecting the neighboring molecule in the tetramer. This suggests an allosteric modulation affecting the enzyme’s activity and explaining why citrate, which mimics ATP binding, was bound only in one subunit. However, this hypothesis requires further kinetic and structural analyses to show a potential regulatory mechanism.

In previous studies, *Pf*HK was kinetically characterized and found to share various similarities with other hexokinases. The products G6P and ADP regulate the enzyme via a feedback mechanism, and ATP was shown to inhibit its activity [7]. The *Pf*HK K*_M_* value for glucose obtained in this study was lower than those obtained in previous studies [7,12]. Interestingly, the K*_M_* for glucose measured for *Cryptosporidium parvum* and *Toxoplasma gondii* hexokinase (*Cp*HK, *Tg*HK) were similarly low or even lower (Table 2). A lower K*_M_* for glucose suggests low glucose concentrations within the parasite and is crucial for its survival [42,43]. However, one should note that there are minor deviations concerning the assay conditions, for example, the pH values, which range from 7.0 to 7.9 in different studies, which could also lead to changed enzymatic characteristics. It has already been shown that in infected erythrocytes, hexokinase activity and glucose consumption are equally significantly increased. Consequently, this increase affects the levels of reduced glutathione (GSH) by providing NADPH for downstream redox reactions, leading to a 25-fold increase in the production of GSH [12].

Interfering with the parasite’s antioxidative defense system is characteristic not only of drugs currently used in clinical practice but also of novel strategies of drug development [44,45,46,47]. Within this study, we showed for the first time that *Pf*HK is inhibited upon both S-glutathionylation and S-nitrosation. Probably, only 10–11 cysteines are accessible within the three-dimensional structure. Cysteines near the active site (C77, C85, C186, C273) or cysteines within the tetrameric interface (C193, C346) seem likely to be modified, induce conformational changes, and thus impair proper catalysis. Interestingly, C193 is highly conserved among *Plasmodium* species, whereas C346 is restricted to *P. falciparum* and six other species infecting chimpanzees and gorillas (Appendix A). As already mentioned, C249 and C260 would allow for the formation of a disulfide but were reduced in the present structure, most probably due to the presence of DTT in the crystallization buffer. This disulfide would not only be more susceptible to oxPTMs and subsequent enzymatic regulation but could also be the target for redox-active proteins. Indeed, *Pf*HK was found to be a target of thioredoxin, glutaredoxin, and the *Plasmodium*-specific plasmoredoxin, further indicating its importance in redox regulatory processes within the parasite [48].

Besides their important role in glycolysis, hexokinases are reported to be moonlighting proteins, which is a term for proteins performing different independent functions. These functions include the regulation of redox signaling or performing as a gene repressor, protein kinase, or immune receptor. It is known that mammalian, yeast, plant, and numerous parasitic hexokinases are post-translationally modified as a response to environmental changes [49,50]. Many hexokinases are known to be S-nitrosated, and S-nitrosation was also predicted for *Pf*HK [18]. Depending on the organism studied, S-nitrosation can either inhibit hexokinase activity or does not affect the enzymatic activity. Also, the incubations with HK substrates can have different effects: incubations with glucose prior to the modification prevented inhibition, whereas incubations with ATP could either prevent or increase inhibition. An increase was possibly mediated by conformational changes that increased the exposure of reactive thiols to nitric oxide (NO)-donors [51,52,53].

Numerous studies provide data about the S-glutathionylation of glyceraldehyde-3-phosphate-dehydrogenase (GAPDH) and other glycolytic enzymes that this modification targets, but the effect of S-glutathionylation on hexokinases is scarcely known. S-glutathionylation patterns in *P. falciparum* have already been thoroughly studied and indicate redox regulation in the malaria parasite: GAPDH and pyruvate kinase were shown to be inhibited by S-glutathionylation [17], which is comparable to the present data on *Pf*HK. The present study provides additional data that is consistent with those previous findings.

The fact that *P. falciparum* vitally depends on glucose as an energy supply and that *Pf*HK is putatively and significantly involved in and controlled by redox regulatory processes makes *Pf*HK a very promising drug target by combining two strategies. Moreover, the quaternary organization of *Plasmodium* hexokinases is unique and potentially provides unknown mechanisms of enzymatic activity regulation [15]. Due to this uniqueness, the present structure of *Pf*HK and the novel insights into the mechanism and the potential redox regulation provide a good basis for designing new *Plasmodium*-specific inhibitors.

## 4. Materials and Methods

Unless stated elsewhere, all reagents were obtained from Merck (Darmstadt, Germany) or from Roth (Karlsruhe, Germany) and purchased with the highest purity available.

### 4.1. Heterologous Expression and Purification of Recombinant PfHK

Due to differences in codon usage among *P. falciparum* and *E. coli*, the hexokinase gene sequence was ordered codon-optimized for *E. coli* from General Biosystems (PlasmoDB ID: PF3D7_0624000). The sequence was cloned into pET30a-c(+) with NdeI and XhoII, leading to a C-terminal His_6_-tag. Heterologous expression was performed in C41(DE3) *E. coli* cells. Cells were grown to an OD_600_ of 0.6 and induced with 500 µM IPTG. Expression was performed for 18 h at 25 °C. Cells were harvested at 10,000× *g* and 4 °C for 15 min. After adding a protease inhibitor cocktail (4 nM cystatin, 150 nM pepstatin, 100 µM phenylmethylsulfonyl fluoride), cells were stored in 50 mM HEPES, pH 8.0, 300 mM NaCl, 20 mM imidazole, and 10% glycerol at −20 °C until lysis. Cells were lysed with 1 mg DNase, 16 mg lysozyme, 30% glycerol, and 2% triton X-100 with a total volume of 50 mL per liter of *E. coli* culture, which was adjusted with a storage buffer. Lysis was conducted overnight for at least 14 h at 4 °C. After three cycles of sonication (30 s, 4 °C, 70% power), cells were centrifuged (30 min, 25,000× *g*, 4 °C). The supernatant was applied to a Talon^®^ metal affinity resin (Clontech, Heidelberg, Germany) using 1 mL resin per liter of *E. coli* culture (1 column volume = CV). To exclude non-specifically bound proteins, the column was washed with a 20 CV storage buffer before eluting hexokinase with 50, 100, and 200 mM imidazole at 4 CV, respectively. Eluted fractions were checked for purity with SDS-PAGE and 12% acrylamide and subsequently concentrated with a Vivaspin 15R centrifugal concentrator (Sartorius, Goettingen, Germany; MWCO 30,000 Da) to a final concentration of ~3 mg/mL or ~10 mg/mL for crystallization trials. The final concentration was measured with a Biospectrometer^®^ basic (A_280_, ε_PfHK_ = 74,237.5 M^−1^ cm^−1^, Eppendorf, Hamburg, Germany). Size exclusion chromatography was performed with a HiLoad 16/60 Superdex^TM^ 200 prepacked column connected to an ÄKTA^TM^ pure protein purification system (Cytiva, Freiburg im Breisgau, Germany) to assess the oligomerization behavior of *Pf*HK.

### 4.2. Kinetic Characterization of PfHK

The activity of *Pf*HK was measured by observing the increase of NADPH at 340 nm at 25 °C in an assay coupled to glucose-6-phosphate dehydrogenase (G6PD). The standard assay contained 15 mM D-glucose, 4 mM ATP, 1 mM NADP^+^, and 5 U/mL G6PD (from baker’s yeast, Merck, Darmstadt, Germany) in 100 mM triethanolamine, with a pH of 7.5 and 10 mM of MgCl_2_. To determine Michaelis–Menten constants, ATP was used in concentrations ranging from 0.1 to 5 mM; D-glucose was used in concentrations ranging from 0.02 to 30 mM. The data were analyzed using GraphPad Prism 8 (see also ***‘Data analyses and statistics’***). Each value is a mean ± SD from at least three independent determinations using different biological batches of enzymes.

### 4.3. Crystallization

Crystals were grown by the sitting drop technique, using 20% PEG 3350, 200 mM sodium citrate, and a 0.1 M sodium citrate buffer, with a pH ranging from 3.4 to 4.0. The protein solution contained 5–8 mg·mL^−1^ *Pf*HK, 10 mM glucose, 10 mM ATP, and 5 mM DTT. As a cryoprotectant, 25% ethylene glycol was used.

### 4.4. Data Collection and Refinement

All diffraction data were recorded at 100 K by using synchrotron radiation at the Swiss Light Source in Villigen, Switzerland (beamline X10SA, Detector: EIGER). Prior to data collection, the crystals were soaked in mother liquor containing 25% ethylene glycol. The data sets were processed with XDS [54], which included routines for space group determination. The trigonal crystals diffracted up to 2.8 Å.

The structure was solved via molecular replacement methods. The search model was generated via homology modeling with SWISS-MODEL [55]. The open (PDB ID: **6VYF**) and closed (PDB ID: **6VYG**) forms of HK from *P. vivax*, which had 89.6% sequence identity to *Pf*HK, were used as templates. The correct space group (P3_1_21, with two monomers in the asymmetric unit) could be determined via molecular replacement searches using the closed form of *Pv*HK. The initial R_free_ was 47.8%, which dropped to 28% after several rounds of refinement and manual rebuilding. The final statistics are shown in Table 1. Molecular replacement and structure refinement were carried out with the PHENIX program suite [56,57], and the interactive graphics program Coot [58] was used for model building. Manual rebuilding and subsequent refinement resulted in a PfHK model, comprising residues I25 to N486, with one glucose and one citrate molecule in subunit A, and residues I23 to I489, with one glucose molecule in subunit B. The structure was well defined by the 2fofc electron density, except for some side chains at the surface and residues G134 to S137 of subunit B.

Superimposition of the structures was performed with the SSM algorithm tool [59] in the Coot graphics package. The SSM algorithm tool is a structural alignment tool based on secondary structure matching. Molecular graphics images were produced using the UCSF Chimera package [60].

### 4.5. Gel Filtration Chromatography

To remove excess chemicals like DTT, GSSG, or *S*-nitrosocysteine (Cys-NO), gel filtration chromatography was conducted. For this purpose, P6 gel powder (Biorad, Feldkirchen, Germany) with a fractionation range from 1000 to 6000 Da was soaked for at least 1 h in *Pf*HK storage buffer. Afterwards, 2 mL columns were loaded with the pre-soaked gel, centrifuged at 1000× *g* for 2 min at 4 °C, and the flow-through was discarded. To achieve efficient desalting of protein samples, the remaining volume of gel after centrifugation should be used in a 5-fold excess over the protein volume. Protein samples were applied, and the columns were centrifuged again to obtain a desalted protein in the flow-through. After each desalting step, the protein concentration was determined again.

### 4.6. Glutathionylation

To ensure that all cysteines were reduced, *Pf*HK was incubated with 5 mM DTT for 30 min at 4 °C. Excess DTT was removed via gel filtration chromatography, as described above. Afterwards, 1 mg·mL *Pf*HK was incubated with increasing concentrations (0 to 10 mM) of oxidized glutathione (GSSG) at 37 °C for 10 min. To stop the reaction and remove excess GSSG, the samples were immediately desalted after the respective incubation and kept at 25 °C until used for Western blot or enzymatic analysis. To investigate the time course of glutathionylation-mediated inhibition, 5 mM GSSG was used for incubations, and an aliquot was taken out to assess the enzymatic activity at different time points (0, 5, 10, 20, and 30 min). To validate glutathionylation with Western blot analyses, protein samples were prepared with a sample buffer and directly applied to SDS-PAGE followed by semi-dry Western blot analysis with a glutathione antibody (Abcam, Dresden, Germany; Glutathione antibody [D8] (ab19534); diluted 1:250 in TBST).

### 4.7. Synthesis of S-Nitrosocysteine, Nitrosation, and Biotin-Switch Assay

As already described for glutathionylation, *Pf*PK was reduced by adding DTT, which was removed via gel filtration chromatography. The nitrosation of protein samples was induced via incubations with CysNO, which was prepared separately for each experiment. All solutions were freshly dissolved and kept on ice and in the dark to avoid loss of concentration due to decreased stability. To synthesize 95.24 mM CysNO, 250 mM sodium nitrite (Roth, Karlsruhe, Germany) was incubated with 250 mM L-cysteine-hydrochloride-monohydrate and 1 M HCl in a 4:4:1 ratio for at least 5 min. Before starting the nitrosation, the CysNO solution had to be neutralized with 1 M NaOH (ratio 6:1) to a final pH of 7.5. The final CysNO solution had an orange-red color. To study nitrosation, 1 mg/mL pre-reduced *Pf*HK was incubated with different concentrations of CysNO, ranging from 0 to 5 mM or 1 mM for time-course experiments. The incubations were conducted at 25 °C in the dark. To stop the reaction and investigate the enzymatic activity, samples were desalted. If the samples were analyzed via Western blot, the reaction was stopped by adding 100% ice-cold acetone followed by protein precipitation for 30 min at −20 °C. Samples were applied to a biotin-switch assay, which could indirectly detect the presence of nitroso groups within protein samples and was previously described in detail [18]. To describe the assay in brief, the samples were washed with 70% ice-cold acetone to remove excess CysNO, followed by blocking unmodified cysteines with 0.2 M iodoacetamide in a blocking buffer (8 M urea, 50 mM Tris pH 8.0, 1 mM EDTA, 0.1 mM neocuproine) for 1 h at 50 °C to alkylate the thiols. Again, the incubation was stopped by adding acetone, conducting protein precipitation at -20 °C, and performing three washing steps. During the next incubation, conducted in a labeling buffer (4 M urea, 50 mM Tris pH 8.0, 1 mM EDTA, 0.01 neocuproine), the denitrosation of thiols was achieved by adding 20 mM of sodium ascorbate. Simultaneously, the denitrosated thiols were labeled with 0.2 mM iodoacetyl-PEG2-biotin for 1 h at 25 °C in the dark. Samples without sodium ascorbate served as negative controls in addition to samples without CysNO. The reaction was again stopped with acetone and, after protein precipitation and the washing steps, protein pellets were resuspended in a blocking buffer and SDS-PAGE sample buffer. Finally, semi-dry Western blot analysis with a biotin antibody (Santa Cruz, CA, USA; Biotin Antibody (33): sc-101,339; diluted 1:1000 in 5% nonfat milk in TBST) was performed to detect nitrosation.

### 4.8. Mass Spectrometry Analysis

To identify cysteines that are sensitive to oxPTMs, reduced PfHK was incubated with varying concentrations of GSSG and Cys-NO, as described above. The MALDI-TOF MS analyses were performed at the Core Facility of Mass Spectrometry and Elemental Analysis, Philipps University Marburg (timsTOF Pro (Bruker), software: PEAKS 8.5, Bioinformatics Solutions Inc., Waterloo, ON, Canada).

### 4.9. Data Analyses and Statistics

To kinetically evaluate modified *Pf*HK and calculate Michaelis–Menten constants and V_max_, we used Microsoft Excel and GraphPad Prism 8. The elution profile of size exclusion chromatography was analyzed with Unicorn^TM^ 7.0, the control and evaluation software of the ÄKTA^TM^ system, also provided by Cytiva. Chemiluminescent imaging of Western blot analysis was conducted with ChemoStar ECL and a fluorescent imager (Intas, Göttingen, Germany).

## Figures and Tables

**Figure 1 ijms-24-12739-f001:**
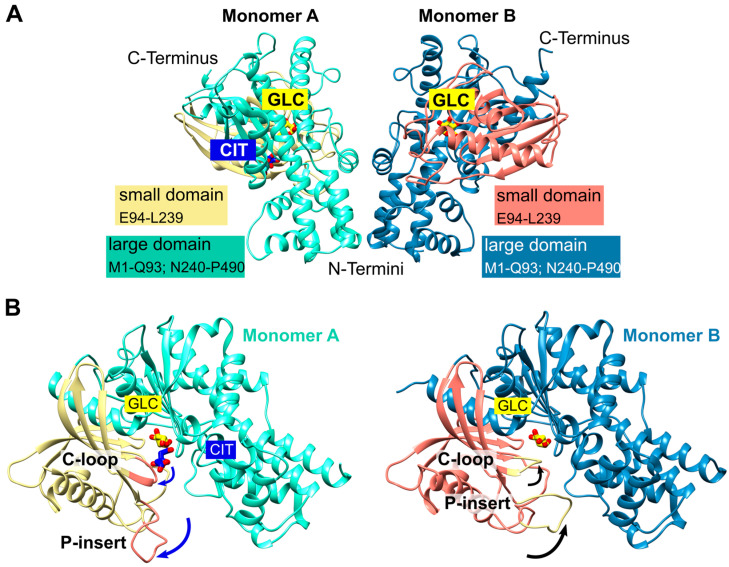
*Pf*HK overall fold. (**A**) The asymmetric unit of *Pf*HK contained two monomers. Each monomer consists of two domains: a small domain (E94-L239) and a large domain (M1-Q93; N240-P490). (**B**) *Pf*HK monomer A was complexed with glucose (GLC) and citrate (CIT), whereas, in monomer B, only GLC was bound. Superimposition of both *Pf*HK monomers revealed a shift of two loops: the catalytic loop (C-loop: residues G104-F108) and a *Plasmodium*-specific insertion (P-insert: residues L130-G141), most likely evoked by the binding of citrate within monomer A.

**Figure 2 ijms-24-12739-f002:**
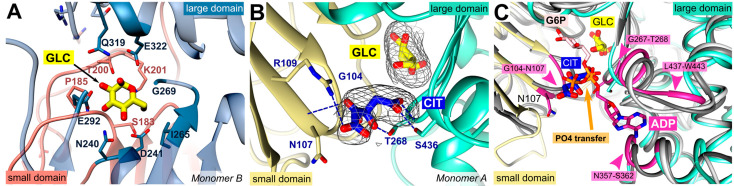
*Pf*HK monomers provided information about the highly conserved substrate binding sites. (**A**) Glucose (GLC) was bound within the active site of *Pf*HK monomer B and coordinated by the highlighted residues of both domains (small domain = salmon; large domain = steel blue) that are conserved among different species. (**B**) Close-up of the active site of *Pf*HK monomer A with GLC and citrate (CIT) bound. Citrate was hydrogen-bonded by residues G104, N107, T268, and S436. The final 2fo-fc map, covering the substrates GLC and CIT, is contoured at 2.0 σ. (**C**) *Pf*HK monomer A was superimposed with two different human hexokinase structures (PDB IDs: **1HKB**, dark grey; **1DGK**, light grey) to identify the corresponding glucose-6-phosphate (G6P) and ADP/ATP binding sites in *Pf*HK. The ADP/ATP binding site is highlighted in violet. PO_4_, phosphate.

**Figure 3 ijms-24-12739-f003:**
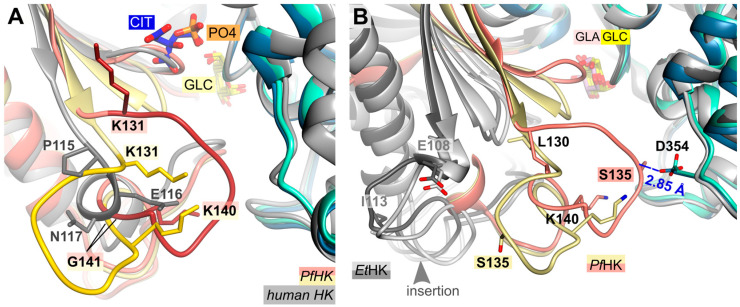
Structural comparison with hexokinases from *H. sapiens* and *E. tenella*. (**A**) *Pf*HK revealed a *Plasmodium*-specific insertion (P-insert, residues K131-K141) when superimposed with human hexokinase (PDB ID: **1DGK**, grey). The human hexokinase structure was complexed with glucose (GLC) and phosphate (PO_4_). (**B**) *Pf*HK monomers A and B were superimposed with two different *Et*HK structures (PDB IDs: **6KSR**, dark grey; **6KSJ**, light grey) to compare the P-insert and a similar insertion from *Et*HK. *GLA*, *galactose*.

**Figure 4 ijms-24-12739-f004:**
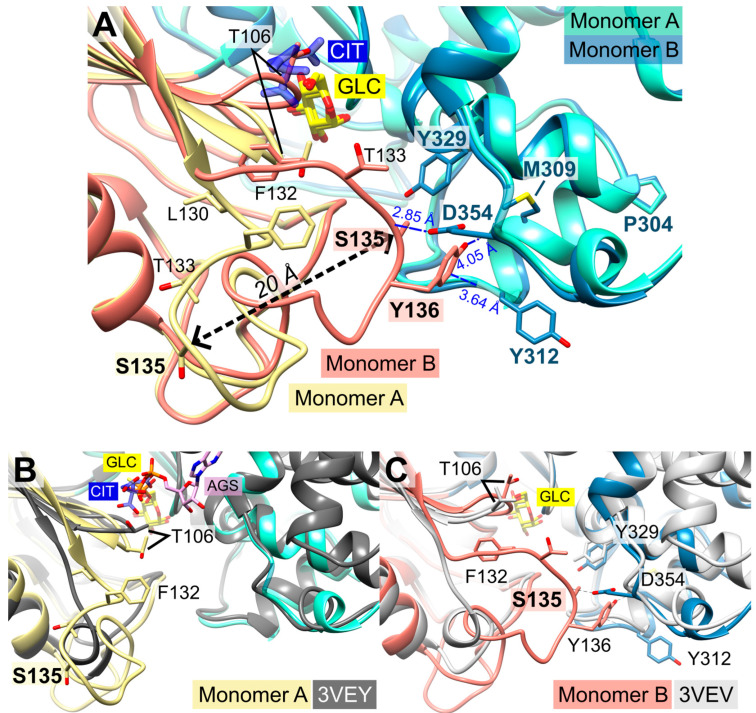
Conformational changes of the small domains induced by the binding of citrate. The catalytic loop is indicated by T106. (**A**) Detailed view of the superimposed *Pf*HK monomers A and B shows the 20 Å shift of residue S135 induced by binding citrate (CIT). In the tetrameric assembly, residues P304-Y312 are part of the AB interface. (**B**) Superimposition of *Pf*HK monomer A with human glucokinase complexed with glucose and ATP analog ATPγS (AGS, PDB ID: **3VEY**). (**C**) Superimposition of *Pf*HK monomer B with human glucokinase complexed with glucose (GLC) (PDB ID: **3VEV**).

**Figure 5 ijms-24-12739-f005:**
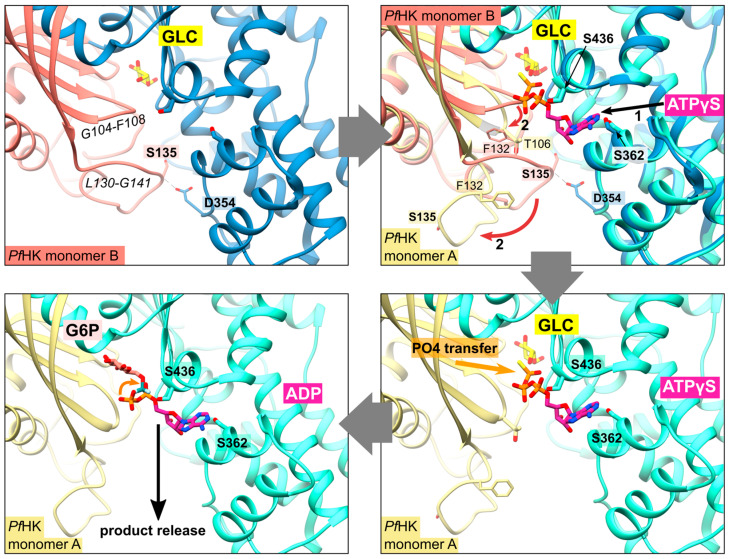
Proposed mechanism upon substrate binding and product release. After the binding of glucose (GLC), the binding of ATP evokes a conformational change of resides G104-F108 and L130-G141 (shown via superimposition of human glucokinase with bound ATPγS, PDB ID: **3VEY**, with *Pf*HK monomers A and B). This broadens the active site and thereby facilitates the release of the products glucose-6-phosphate (G6P) and ADP after the transfer of phosphate.

**Figure 6 ijms-24-12739-f006:**
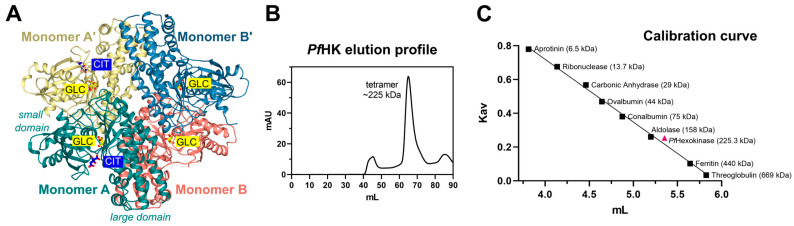
Tetrameric assembly of *Pf*HK. (**A**) *Pf*HK tetramer created with a crystallographic symmetry operation. (**B**) *Pf*HK was purified via Talon^®^ metal affinity chromatography followed by size exclusion chromatography using an ÄKTA^TM^ pure protein purification system, Superdex^TM^ 200 column. *Pf*HK eluted at ~65 mL. (**C**) Size exclusion chromatography calibration curve for *Pf*HK. The estimated molecular mass is indicated in the trend line as a pink triangle, corresponding to ~225 kDa (*Pf*HK monomer = 56.3 kDa, expected *Pf*HK tetramer = 225 kDa).

**Figure 7 ijms-24-12739-f007:**
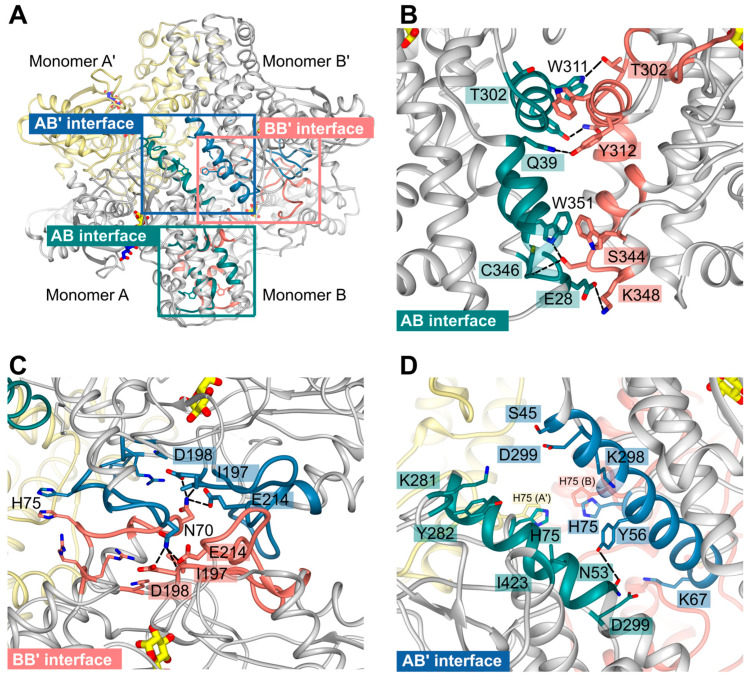
*Pf*HK unique tetrameric interfaces. (**A**) The *Pf*HK tetramer has four interaction sites that are highlighted with a blue (AB′), salmon (AA′, BB′), and green square (AB), respectively. (**B**) The AB interface is strongly connected via seven hydrogen bonds and one additional salt bridge between E28 and K348. Mainly residues from the large domains form the interface. (**C**) Mainly residues from the small domains connect monomers AA′ or BB′. N70 from each monomer could be hydrogen-bonded to regions L196-D198 and E214. In AA′, there is an additional hydrogen bond between R207 and K283 (not shown). (**D**) The AB′ interface is formed via hydrophobic interactions between two helices and a hydrogen bond between N53 and Y56. Distances between the H75 sidechains range from 3.6 to 5.9 Å. Hydrogen bonds are indicated by black dashed lines.

**Figure 8 ijms-24-12739-f008:**
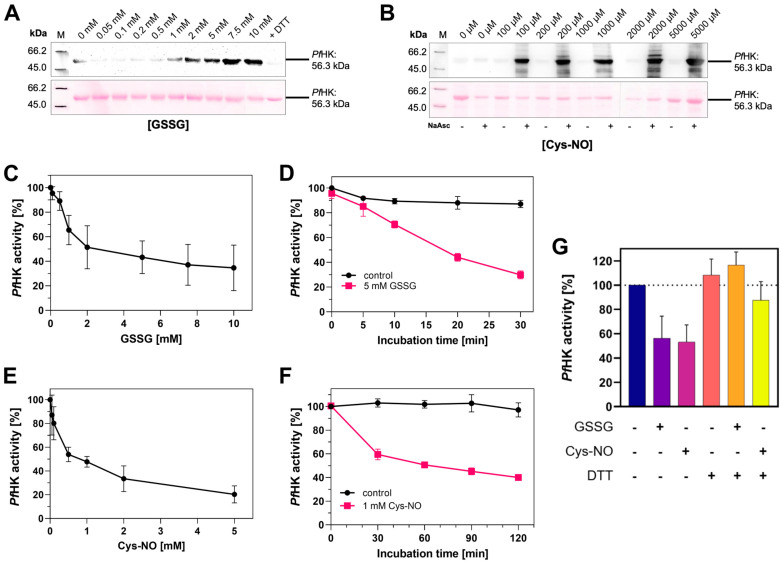
*Pf*HK is inhibited via S-glutathionylation and S-nitrosation. (**A**) S-glutathionylated samples from (**C**) were analyzed via Western blot. S-glutathionylated *Pf*HK was detected with a glutathione antibody. SDS-PAGE loading buffer with DTT served as a chemical negative control. (**B**) S-nitrosated samples from (**E**) were analyzed via Western blot. S-nitrosated *Pf*HK was detected with a biotin antibody. Samples without NaAsc (sodium ascorbate) served as a chemical negative control. In both (**E**) and (**F**), ~2 µg protein per lane was used and Ponceau staining served as a loading control. (**C**) Reduced *Pf*HK was incubated with 0–10 mM GSSG for 10 min at 37 °C. The activity of the sample incubated with 0 mM GSSG (control) was defined as 100%. (**D**) Reduced *Pf*HK was incubated with 0 mM (control) or 5 mM GSSG for 0–30 min at 37 °C. The activity of the control sample at 0 min was defined as 100%. (**E**) Reduced *Pf*HK was incubated with 0–5 mM S-nitrosocysteine (Cys-NO) for 60 min at 25 °C. The activity of the sample incubated with 0 mM Cys-NO (control) was defined as 100%. (**F**) Reduced *Pf*HK was incubated with 0 mM (control) or 1 mM Cys-NO for 0–120 min at 25 °C. The activity of the control sample at 0 min was defined as 100%. Values from (**C**–**F**) are shown as mean values ± SD. Measurements were performed in at least three biologically independent triplicates. (**G**) Incubation of S-glutathionylated and S-nitrosated *Pf*HK with DTT for 30 min at 4 °C restored its enzymatic activity. Values are shown as mean values ± SD. Measurements were performed in at least three biologically independent triplicates.

**Table 1 ijms-24-12739-t001:** Data collection and refinement statistics.

PDB Accession Code	7ZZI (*Pf*HK)
Space group	P3_1_21
**Unit cell parameters**	
a, b, c (Å)	125.24, 125.24, 120.93
α, β, γ (°)	90, 90, 120
**Data collection**	
Beamline	SLS beam line X10SA
Temperature (K)	100
Resolution range	49.5–2.8 (2.9–2.8)
Wilson B-factor Å^2^	95.8
Total reflections	278,108 (26,474)
Unique reflections	27,459 (2644)
Multiplicity	10.1 (10.0)
Completeness (%)	99.7 (97.1)
Mean I/σ (I)	13.1 (0.8)
Wilson B-factor	95.8
R-merge (%)	0.12 (3.41)
R-pim (%)	0.04 (1.12)
CC1/2 (%)	0.999 (0.305)
Molecules per ASU	2
**Refinement**	
Reflections used in refinement	27,456 (2644)
Reflections used for R-free	2746 (264)
R-work	0.199 (0.428)
R-free	0.268 (0.481)
Protein residues	930
Macromolecules average B-factor Å^2^	89.9
Ligands average B-factor Å^2^	82.8
Solvent average B-factor Å^2^	82.2
**RMS deviations**	
Bonds (Å)	0.01
Angles (°)	1.17
Ramachandran favored (%)	96.11

Statistics for the highest-resolution shell are shown in parentheses.

**Table 2 ijms-24-12739-t002:** K*_M_* values of hexokinases from different studies and species.

	K*_M_* (mM)	
	Glucose	ATP
*Pf*HK, present study	0.13 ± 0.01	0.83 ± 0.08
*Pf*HK, Harris et al. 2013 [7]	0.62 ± 0.06	0.66 ± 0.08
*Pf*HK, Roth et al. 1987 [12] *	0.431 ± 0.021	3.1 ± 1.4
*Cp*HK, Yu et al. 2015 [43]	0.138	0.673
*Tg*HK, Saito et al. 2002 [42]	0.0080 ± 0.0008	1.05 ± 0.25

* From infected erythrocytes.

## Data Availability

Coordinates and measured reflection amplitudes have been deposited in the Worldwide Protein Data Bank RCSB PDB (http://pdb.org): code **7ZZI** for *Pf*HK.

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
