# Peer review of "Structural Analysis of Plasmodium falciparum Hexokinase Provides Novel Information about Catalysis Due to a Plasmodium-Specific Insertion"

_ijms, 2023, doi:10.3390/ijms241612739_

Round 1

Reviewer 1 Report

The paper treats the hexokinase from Plasmodium falciparum. Related hexokinases have been reported from Plasmodium vivax in opened and closed states without ligand.  The new structure of PfHK is in the closed state and contains either glucose or glucose and a citrate, which mimics the diphosphate of ADP. The overall structure is similar to the known Plasmodium hexokinase structure with an rmsd of only 1.1 A for 413 residues.
New is the conformation of the P-insert that is specific for Plasmodium hexokinases, but was disordered in the Plasmodium vivax structures. The two observed conformations of the P-insert were related to states before catalysis and for product release. This is derived from superposition with structures with various ligands which were already assigned to pre-catalytic, catalytic and post-catalytic states. The assignment of the P-insert conformations is new. The model of conformational states in the catalytic cycle is not new [30].

Glutathionylation and S-nitrosation were reported for hexokinases before. Here, they were measured for PfHK in a dose- and time-dependent manner. Reversal of this modification with a thiol (DTT) was also shown. Site specificity of modification was measured qualitatively by MS, but the site specific effect of the modifications was not detailed.

Overall, the protein chemical and structural studies on the PfHK complement previous studies on HKs. Determination of the structure and effect of the Plasmodium-specific loop is new. Experiments were performed in a reproducible way and documented comprehensibly. Figures are clear. Links to the references should be given repeatedly, if information from the same paper is used at different places in the paper.

Detailed comments:

l. 91: 6VYG for the closed state

l. 113 and following
Table 1:
Rmerge: there is no note a
Rpim:  there is no note b

l. 143: 6VYG for the closed state

l. 201: "are oriented inward the active site": please rephrase (do you mean "are facing the inside of the active site",  "are pointing into the active site"?)

l. 206: when a structure is used, the reference should be given, here: 3VEY [30]

l. 211-214: "C-loop adopts a pre-catalytic conformation ... corresponds to a post-catalytic conformation." This is confusing.

l. 240-241: Why is the interface AA' the same as interface BB'? A and B are related by ncs, not by crystallographical symmetry.

l. 267 should spell "subunit A (CE1)"

l. 277: Why are the distances in line 257 and line line 277 different (5.9 and 5.8 A)? The lines are dashed not dotted.

l. 307: should spell "of purified recombinant PfHK"

l. 375: "single cysteines" do you mean individual or specific cysteines?

l. 495: should spell "prevent or increase"

l. 560: should spell "via homology"

Suppl.
Figure S2: Is it really the neighboring tetramer or is it the same tetramer, but the neighboring asymmetric unit?

Figures and Tables: Several figure captions cross page borders which must be avoided in the final layout.

Only minor editing required.

Some suggestions are given above.

Reviewer 2 Report

The authors of the current manuscript have presented the crystal structure of Plasmodium falciparum hexokinase. This is a very significant contribution to understanding the mechanism of action of this enzyme in the malarial parasite with the aim of targeting it for the development of novel anti-malarial compounds. The study is well-designed, and the results are convincing. The manuscript is acceptable for publication; however, the authors should address the following queries and concerns.

1.     The authors have shown in the crystal structure and through size-exclusion chromatography, that Pf hexokinase is a tetramer. The following queries (a-d) pertain to the role of oligomeric status in catalysis by the enzyme.

a.     Have the authors identified any condition wherein the enzyme switches from tetramer to dimer? Is the oligomerization dependent on the buffer or protein concentration? If the enzyme dimer exists, is the enzyme more active as a tetramer than when it is a dimer?

b.     Does Pf hexokinase co-purify with glucose, glucose 6-phosphate, ADP, ATP, or citrate at the active site? If so, do these ligands promote the oligomerization of the enzyme?

c.     The authors have alluded to the inhibition of phosphorylation activity of hexokinase by Glucose 6-phosphate and ADP. Is the inhibition caused due to the change in the oligomeric state of the enzyme?

d.     Do either of the post-translational modifications, S-glutathionylation or S-nitrosation cause a switch of the oligomeric state?

2.     What is the order of substrate binding in Pf hexokinase? Is glucose binding conditional to the binding of ATP or is it random? Does the order of substrate binding differ among hexokinases from P. falciparumP. vivax, and human?

3.     Does the closed conformation of apo Pf hexokinase pre-exist in equilibrium with the open conformation, and substrate binding drives the equilibrium towards the closed conformation, or is the closed conformation induced only upon ligand binding?

4.     Can the authors predict the rate-limiting step in the phosphorylation process by PfHK, based on the conformational changes induced upon catalysis? Is the rate-limiting step the catalysis or the release of G6P/ADP?

5.     The authors have provided the title to the figure in the PowerPoint slide as ‘Pf HK is inhibited by S-glutathionylation and S-nitrosation’. However, the figure does not appear to show any kinetic data. In the same figure, the authors have alluded to panels C and E, but the panels are not labeled. The panels in Figure 8 of the PowerPoint slide should either be condensed with Figure 8 of the main manuscript or provided as supporting information with a different title for the figure.

6.     In the methods for purification of PfHK, the authors have mentioned that cell lysis was done overnight for 14 hours and 4°C (lines 525-526). This is followed by 3 cycles of sonication for 30 seconds. Usually, sonication is enough to disrupt the cells of any strain of bacteria. The authors should clarify what overnight lysis means.

Listed below are a few typographical errors that the authors need to correct.

a.  Line 392: '....resulting in movements of the polypeptide backbone of 8A'. Replace 'of' with 'by'

b. Line 392: 'Binding glucose induces.......' Change to ' Binding with glucose induces....'

c.  Line 414: ‘instable’ to ‘unstable’

d.     Line 480: ‘probable’ to ‘probably’

e.     Line 529: ‘unspecifically to ‘non-specifically’

f.       Line 560: ‘vial’ to ‘via’

g.     Line 601: ‘PfPK to ‘PfHK’

h.      Line 604: ‘solved’ to ‘dissolved’
